# Effect of emotional factors on purchase intention in live streaming marketing of agricultural products: A moderated mediation model

**Tianming Han, Jing Han, Jia Liu, Weibao Li** *

School of Economics and Management, North China Institute of Aerospace Engineering, Lang Fang, Hebei Province, China

* liweibao042944@163.com

## Abstract

Emotional factors play a crucial role in streaming live marketing of agricultural products. Some literature explored several emotional factors' impact on consumers' purchase intention. Nonetheless, the interaction and integration effects of these factors have received less attention. Based on Consumer Engagement Theory, SOR model and TAM model, the paper constructs a moderated mediation model of the interactivity/presence, trust/resonance and purchase intention under rural sentiment. A quantitative study based on 365 valid samples is conducted to validate this model. The results indicate that interactivity and presence positively impact on consumers' purchase intention, trust and resonance play a mediating role between interactivity/presence and purchase intention separately. Contrary to our expectations, rural sentiment negatively moderates the relationship between interactivity and resonance. Differences of regression results between urban and rural group indicate that the cultural backgrounds of consumers have an impact on their emotional responses in live streaming of agricultural products. The results illustrate the mechanism of emotional factors in consumers' purchase decisions. Overall, this paper reveals the potential of emotional factors and the development of effective marketing strategies to improve agricultural products sales.

## Introduction

E-commerce live streaming is rapid increasing in China, according to CNNIC, as of the end of June 2023, e-commerce live streaming users reached 526 million and accounted for 48.8% of all internet users [1]. The market scale of China's e-commerce live streaming was predicted to be 4.9 trillion yuan in 2023. Consumers' cognition of the interactive, social and entertaining characteristics of live streaming is deepening. Live streaming provides more favorable prices, more intuitive display and higher trust. Therefore, people are more likely to accept the consumption mode in the live streaming room [2]. Moreover, there are more products that are also starting to be sold on the live streaming room, such as agriculture products. According to data published in 2022 by the Ministry of Commerce of China, the number of rural online stores had exceeded 17.3 million, a 6.3% increase from the previous year, among which 5.73

**Data Availability Statement:** All relevant data are within the manuscript and its Supporting Information files.

**Funding:** Doctor Research Foundation of North China Institute of Aerospace Engineering, BKY-

2021-41, Doctor Tianming Han; S&T Program of Hebei, 23557635D, Doctor Weibao Li; Humanities and Social Science Research Project of Hebei Education Department, SQ191007, Doctor Weibao Li.

**Competing interests:** The authors have declared that no competing interests exist.

million were e-commerce live-streaming of agricultural products, accounting for one-third of rural online stores [3]. In the past, agricultural products were traded through Taobao or other e-commerce platforms, and consumers would consider factors such as the text introduction and images of the product page, buyer comments, and photos taken by buyers; the communication between sellers and consumers is mainly occurred after sales, and less information could be conveyed to consumers. Nowadays, the visualization of live streaming and supporting facilities such as logistics supply chain make live streaming of agricultural products become an vital sale channel of agricultural products.

In June 2022, the East Buy, a popular live studio of agricultural products, had changed the traditional shouting style of live streaming into content marketing. The host Mr. Dong always chats with the consumers and talks about his rural life experience, as if they were his friends, which attracted lots of consumers to watch. For example, during one live streaming session, he said: "when you were a child, you always ate the corn mom boiled for you. The corn was sweet and you were young and carefree. . . You felt pure and safe. . . Then, your parents were young and healthy, and your grandparents were always happy to be with you. . . You are not recalling the corn, you are recalling your life." With these emotional words, the sales of corn had increased dramatically. This friendly chat–style live streaming mode increases the benefit of live streaming and shortens the distance between consumers and the host. East Buy' success lies in the sincere communication between the host and the consumers, stimulating the resonance of consumers, and then affecting their purchase intention. It indicates that emotional factors, such as resonance, sentiment, and presence, are very essential to live-streaming marketing, especially for agricultural products.

With the improvement of e-commerce live streaming, researchers have focused on the influence of emotional variables on purchase intention in live streaming of agricultural products. Zhou et al. found the common stimulation of cognitive and emotional social presence in the live streaming would make consumer generate internal state changes of perceived usefulness and trust, and responded to the purchase intention [4]. Scott Robinette found that emotional factors had more and more influence on consumers, so in order to better obtain the corresponding market value and benefit growth rate, enterprises must pay attention to the ability of improving consumers' emotional resonance [5]. Jiang Weijie et al. believed that emotional marketing could arouse consumers' memory, stimulate the emotional resonance of consumers through marketing means, re-stimulate the emotional needs of consumers, and enhance marketing [6]. The virtual ambience of live streaming could arouse consumers' emotional and cognitive responses in some way, and ultimately determine their behavior [7]. Xiao Jun et al. summarized the phenomenon of emotional communication in live streaming by using "algorithmic emotion", which mainly includes pleasant emotion, emotional trust, emotional infection and sharing, moral sentiment and emotional performance [8]. Online retailers could develop emotional strategies to improve platform traffic and sales performance. One of the advantages of agricultural products live streaming lies in the stronger interactivity between hosts and users, which can increase users' sense of experience and immersion, bring real consumption experience to users, and thus stimulate users' purchase intention [3]. Rustic complex could arouse people's desire to buy local specialty [9]. Integrating Rustic complex into the packaging design of local specialty could stimulate more potential consumers, promote the consumption of local specialty, and drive the development of rural economy [10].

Although Scholars has paid attention to emotional factors' (such as trust, resonance, and presence) effect on consumers' purchase intention [4, 6–8], the interaction and integration effect of these factors need to deeply research, especially the effect of rural sentiment in live streaming marketing of agricultural products.

By reviewing corresponding literature, we find Consumer Engagement Theory, SOR model, and TAM model integrated several emotional factors and provided theoretical guidance for our research. Consumer Engagement Theory explained the relationship among cognition, emotion, and behaviors. Through interaction between consumers and products, consumers' experience of the products will be internalized into consumer cognition and emotion, which motivate consumers' purchase-related behaviors [11, 12]. Engagement occurred only after a relationship is formed based on trust and commitment [13]. The Stimulus–Organism–Response (SOR) model indicated that organism internalized the outside stimulus into psychological information and made some behavioral responses. So emotional states (e.g. pleasure, arousal) are significant mediators of purchase intention [14]. Atmospheric cues as a external environmental factor of online retailing, would stimulate consumers' emotional reactions [15]. Technology acceptance model (TAM) explained why consumers are willing to use and shop on e-commerce live streaming [16].

The interactivity and integration of trust, resonance and presence has been proven to affect purchase intention by Consumer Engagement Theory, SOR model, and TAM model, and rural sentiment also has obviously impact on agricultural products marketing. Therefore, this paper will explore that in the live streaming of agricultural products, how four emotional factors——trust, resonance, presence and rural sentiment influence purchase intention. We focus on examining the mediating role of resonance and trust between interactivity/presence and purchase intention, and the moderating role of rural sentiment between interactivity/presence and trust/resonance. This study conducts a moderated mediation model which integrates both mediation and moderation effects that aims to illustrate the impact mechanism of emotional factors on purchase intention. We will develop corresponding measurement scales according to the features of live streaming of agricultural products, then test the aforementioned mediating role and moderating effect through regression analysis.

## Literature review and hypotheses development

### Interactivity and purchase intention

Interactivity relies on the communication among participants synchronously and asynchronously, along with information exchanges [17]. In the e-commerce live content marketing, interactivity interpreted consumers' psychological perception of the time and space information exchange and communication process between themselves and the host when watching live-stream [18], which is a crucial atmosphere sign that can affect users' cognition and emotions [19]. With the real-time communication tools, users can interact with hosts and other users beyond time and space limitations. In addition, online reviews can shorten the interactivity distance between consumers and the host and strengthen consumers' product awareness and purchasing intention through online mutual visits [20]. Live streaming has characteristics such as dynamism and product visibility. The introduction of products and interactive methods by the host can affect consumers' viewing experience and curiosity, and consumers can raise questions through bullet screens, allowing the host to provide timely and accurate answers. This series of interactive processes are favorable to eliminating consumers' doubts and strengthening mutual trust. In addition, the interactive behaviors of other viewers in the live-stream, such as inquiries, positive reviews and placing orders, can also have an impact on the audience, thereby stimulating purchase intention. According this discussion, the hypothesis is proposed:

**H1:** Interactivity in streaming live marketing of agricultural products positively influences consumers' purchase intention.

## Presence and purchase intention

Presence refers to the comprehension of psychological intimacy and physical immediacy between buyers and sellers [21–23]. Psychological intimacy can be created in the frequent conversations between consumers and marketers, in which buyers feel warmth, trust, and resonance [23, 24]. Physical immediacy makes buyers feel like they are present at the sellers' virtual location, even though the sellers are physical distantly [25].

In live streaming shopping of agricultural products, the host's introduction of the products and description of the agricultural scene increased the consumer's perception of presence, making them yearn for the agricultural products and scene, thereby increasing their willingness to buy. In the live streaming of agricultural products, the hosts taste and describe the flavor of products to consumers, and explain the process of planting by farmers and the challenging harvesting processes. These tests and descriptions create a sense of real farmlands, in which consumers feel like that they see and taste these agricultural products personally, therefore enhanced their purchase intention. According this discussion, the hypothesis is proposed:

**H2.** Presence in the live-streaming marketing of agricultural products positively influences consumers' purchase intention.

## Trust as a mediator

**Interactivity, trust, and purchase intention.** Trust, indeed, is built through social interactivity with others and their surroundings [26]. Trust refers to the extend to which consumers have confidence in companies, while the confidence come from consumers' perception obtained from interactivity with company or its products and service [27]. Shu et al verified consumers' trust was more from the interactivity with the host and the atmosphere of the live streaming, namely the interactivity has a significant influence on perceived trust, and the perceived trust can trigger consumers' purchase intention based on health concept [28]. Host identity also affects trust. The county magistrate directly interacts with the masses through new media, and personally promotes agricultural products may make consumers feel his quality and behavior are commendable; moreover, the authority of the identity of the county magistrate endorsements the product quality virtually, which make consumers trust the quality of agricultural products and are more willing to buy [29]. According this discussion, the hypothesis is proposed:

**H3a.** Trust mediates interactivity and purchase intention.

**Presence, trust, and purchase intention.** Higher presence allows consumers to obtain more reliable information from live streaming room, thus enhancing their perceived trust in the products and increasing their purchase intention. Live streaming shopping, unlike offline shopping, face-to-face interactivity between consumers and hosts is not able to achieve because of the temporal and spatial separation [30]. That's why online trust is a crucial factor for live streaming shopping. Online trust refers to a live streaming platform's ability to provide truthful information and deliver on consumers' expectations [31]. In live streaming of agricultural products, the hosts introduce all aspects of information about the agricultural products in orchards, farmland, markets, processing factories and other scenes, this gives a strong visual impression to consumers. The consumers were immersed in the world the hosts had created, and the products seems like right in front within reach. This presence enhanced audiences' trust in the products, which made it easier for buyers to place a order. According this discussion, the hypothesis is proposed:

**H3b.** Trust mediates presence and purchase intention.

## Resonance as a mediator

Resonance is a reflection of collective memory and group emotions [32], and a spiritual need for people to seek sense of collective existence [33, 34]. Keller believed resonance is process from mental to behavior [35]. In the live streaming platform, atmospheric resonance and emotional resonance are two categories of resonance [36]. Atmospheric resonance refers to attractive scenes, sounds, pictures, and words, etc. in live streaming room [37, 38]. Emotional resonance refers to consumers' emotional needs and empathy obtained through interactivity with hosts in live streaming room [39]. Technology brings atmospheric resonance on sensory, while interactivity brings emotional resonance on psychology. Both technology and interactivity stimulate consumers' emotional reactions, and enhance their purchase intention.

**Interactivity, resonance, and purchase intention.** Resonance, the synonyms of empathy, means sensing the sensations felt by others [40]. Deng et al assume that in the live streaming, the closeness of the host and the values conveyed to consumers can trigger the emotional resonance of users in various spiritual fields [39]. Consumers often exchange their experience of online shopping and evaluate goods and services. This interactivity is mutual and non-utilitarian, which is easy to cause the resonance between consumers [41]. The accumulation of resonance will have a lasting impact on psychological and physical health and affect people's behavior [42]. Watching narrative advertisements is also a way for businesses to communicate with consumers. When watching narrative advertisements, consumers can perceive the similarity between themselves and the advertising characters, and produce emotional resonance. This emotional resonance experience generated by the similarity can further improve consumers' purchase intention [32]. Meanwhile, the consumers who watches live streaming and interact frequently, will feel strong belonging, which causes the subtle emotional resonance between the audience and internet celebrity and arouse the audience's inner self, make them completely immersed in live streaming. Not only does they have a sense of identity with internet celebrity and their recommended products, but also enhance purchase intention [43]. According this discussion, the following hypothesis is proposed:

H4a. Resonance mediates interactivity and purchase intention.

**Presence, resonance, and purchase intention.** Higher sense of presence helps to establish closer relationships between consumers and hosts in live streaming shopping, to arouse consumers' psychological emotions and ultimately influence their intentions and behaviors. Under the catalysis of emotions and resonance, a sense of presence can enhance consumers' purchase intention. Specifically, presence increases the consumers' atmosphere resonance through shaping all kinds of agricultural scenes (i.e. the orchards, farmland, markets, processing factories, et al.). Scenes can provide consumers with more information, and can even replace language communication and enhance emotional connection to a certain extent; Sounds, pictures, and hosts' narration (such as stories, history, and poetry about agricultural products on sale) is extremely touching and can stimulate purchase intention [44]. On the other hand, presence enhances emotional resonance of consumers through the hosts' introduction to agricultural products. The hosts, with friendly and close images, resonate with consumers, enhancing their perceived value and stimulating their recognition of the live streaming room, and finally encourage consumers to pay for their emotional needs and empathy [34]. consumers interact in a common sense of presence to generate emotional connections, forming a high degree of group emotional resonance, thereby creating significant stickiness to the product or brand. According this discussion, the hypothesis is proposed:

**H4b.** Resonance mediates presence and purchase intention.

## Rural sentiment as a moderator

Grain is the foundation for the survival and development of a nation. "Agriculture, rural areas, and farmers" matters a lot. The rural sentiment can be understood as the nostalgia and love of people with or without rural life experience for the events related to "agriculture, rural areas and farmers". Sentiment is like a mirror of the past, carrying the common memories of the target group [45]. With consumers' nostalgic preferences, sellers use similar slogans or packaging to evoke consumers' good memories and brand resonance [46]. Reasonably utilize of sentiment can produce good marketing effects in live streaming. In particular, hosts can start with the memories of rural life when selling agricultural products, stimulating consumers' rural sentiment. Consumers' rural sentiment can create a "social presence" through interacting with hosts and other consumers in the live stream, so that they could get more product information and their emotional resonance would be more authentic [47], and increasing the trust of the host.

In addition to interactivity in live streaming, hosts also create physical scenes to arouse consumers' rural sentiment. Live streaming in orchards, farmland, markets, processing factories and other scenes give a strong visual impression to consumers and create a "telepresence". Consumers who lived in rural areas when they were young will be aroused their rural sentiment, and enhance their intentions to purchase agricultural products on sale in live streaming [47]. He Jianxun et al. believed that ensuring the perceived quality of products and convey the constant quality of products to nostalgic consumers through advertising, will bring trust to consumers and enhance purchase intention [46]. According this discussion, the hypothesis is proposed:

**H5a.** Rural sentiment moderates the interactivity and trust: for consumers with strong rural sentiment, the positive effect between interactivity and trust is stronger.

**H5b.** Rural sentiment moderates interactivity and resonance: for consumers with strong rural sentiment, the positive effect between interactivity and resonance is stronger.

**H5c.** Rural sentiment moderates the presence and trust: for consumers with strong rural sentiment, the positive effect between presence and trust is stronger.

**H5d.** Rural sentiment moderates the presence and resonance: for consumers with strong rural sentiment, the positive effect between presence and resonance is stronger.

## Mediation model with moderation

As H5a proposed, rural sentiment moderates interactivity and trust, while trust mediates interactivity and purchase intention(H3a). So we propose that rural sentiment might moderate the trust' mediating effect. Similarly, rural sentiment might moderate the mediating effect of resonance. Mazaheri points out that pleasure and arousal will affect consumers' perception of the atmosphere of the e-commerce platform, and then affect the consumers' purchase intention [48]. We conduct a moderated mediation model which integrates the mediating effects of trust/ resonance and the moderating effects of rural sentiment. According this discussion, the hypothesis is proposed:

H6a. Rural sentiment moderates the mediating effect of trust between interactivity and purchase intention positively.

H6b. Rural sentiment moderates the mediating effect of resonance on the relationship between interactivity and purchase intention positively.

H6c. Rural sentiment moderates the mediating effect of trust on the relationship between presence and purchase intention positively.

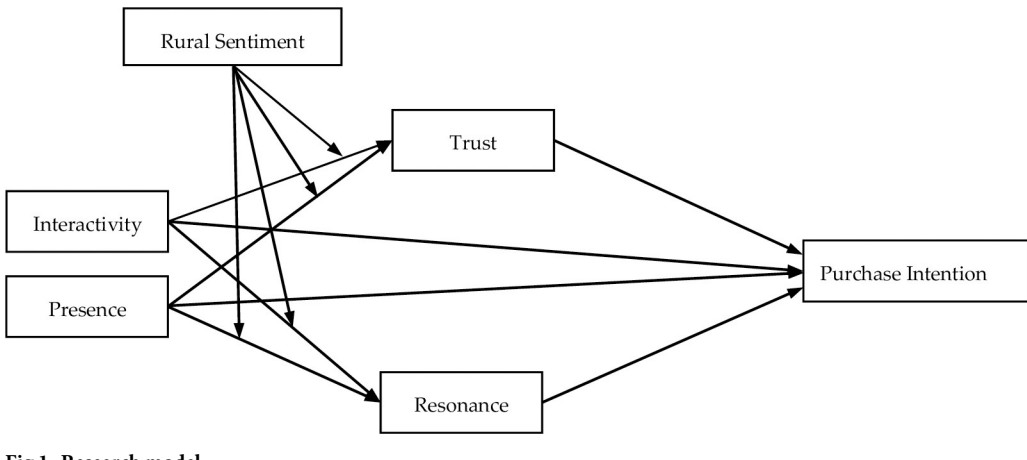

**Fig 1. Research model.**

H6d. Rural sentiment moderates the mediating effect of resonance on the relationship between presence and purchase intention positively.

This study constructs a research model based on the aforementioned research assumptions, as presented in Fig 1.

## Materials and method

### Research subjects and collection procedures

The research subject of this study is e-commerce live streaming users, however, we cannot directly obtain a sampling frame for e-commerce live streaming users. Meanwhile, any netizen is a potential live streaming user. Therefore, our research targets the netizen and sets the question "Have you ever watched an agricultural product e-commerce live streaming" in the questionnaire for screening to obtain our research sample. Potential interviewees were invited to answer the survey questionnaire, excluding minors. All interviewees were clearly informed about the study's purpose and personal privacy before the survey. We mainly collect consumers' attitude data related to streaming live marketing. Interviewees' personal data includes gender, age, income, location, etc. The questionnaire is anonymous and does not collect personal privacy data such as names and phone numbers. Interviewees can withdraw from the survey at any time. This study belongs to investigation research and does not involve human biomedical research. The likelihood and degree of risk posed by this study to the interviewees are not greater than the risks of daily life or routine psychological testing. The research process complies with ethical requirements. This study was approved for exemption from ethics review, and the requirement to obtain informed written consent was waived by the ethics review board.

In order to improve the sample's representativeness, quota sampling method was adopted in this survey, with a planned total sample size of over 1000. According to the 52nd Statistical Report by CNNIC, the urban-rural ratio of Chinese netizen was 72.1%: 27.9%, and the ratio of men to women was 51.4%: 48.6%. In terms of age structure, the proportion of netizen under 20, 21–30, 31–40, 41–50, and over 50 years old was 17.7%, 14.5%, 20.3%, 17.7%, and 29.9%, respectively. According to the structure of Chinese netizen by CNNIC, the urban-rural ratio of respondents in the total sample size was set to 70%: 30%; Set the gender ratio to 50%: 50%; As our research subjects do not include those under 16 years old, the age structure ratio is set to 10%: 25%: 25%: 25%: 15%. The data collection work began on November 27, 2023. Due to our

use of https://www.wjx.cn to collect data, it is difficult to strictly control the quota. Therefore, we calculate the composition ratio of the collected samples in real-time, and accept a deviation of no more than 5% between the sample composition ratio and the quota plan. We plan to end the survey when our total sample size reaches 1000 or more, and the total sample structure meets the quota ratio. Once the sample size is sufficient, we extend this survey to December 26, 2023, the investigation last for one month. A total of 1006 questionnaires with potential interviewees. All interviewees come from a wide range of sources, covering 29 provinces and cities in Chinese Mainland except Xizang.

714 respondents have watched live streaming of agricultural products. This study focus on the immediate psychological feelings of consumers during live streaming, so the daily feelings of users towards live streaming marketing is not suitable as a research object. Therefore, we use a certain experience of consumers watching live-stream as an appropriate research object. Thus, another question "Have you watched agricultural product streaming live marketing more comprehensively in the past two weeks? (The following questions require you to recall the most impressive live-stream in the past two weeks.)" was set in the questionnaire. The interview would quit when respondents answered no. Otherwise, respondents would complete the following questions. There were 391 questionnaires with complete answers to the scale items. After data review, questionnaires that did not have logical responses and were filled out randomly (with an online response time of less than 1 min) were considered invalid and further eliminated, finally 365 valid samples were ultimately obtained. The samples have good representativeness, Table 1 displays the details of sample structure.

## Measurement

Because live streaming of agricultural products involves both agricultural products and immediacy, there is a lack of directly usable scales for measuring the aforementioned variables. This study draws on measurement scales from multiple studies and makes appropriate revisions. In the process of revising and supplementing the scale, measurement items are mainly obtained through group brainstorming and expert opinion methods. A pretest of the questionnaire was performed, and the scale was tested through exploratory factor analysis (EFA), confirmatory factor analysis (CFA) and internal consistency coefficient. After further purification, the final measurement scale was formed. Table 2 presents the constructs, observed indicators and references.

The measurement of rural sentiment (RUS) was developed by referring to the scale proposed by Ma [49], Su [50], Yin [51] and Zheng [52]. There are four measurement items (RUS1, RUS2, RUS3 and RUS4), including "I believe that the long-standing and splendid Chinese agricultural culture needs to be passed down." A Likert scale was used, with 1 = strongly disagree and 7 = strongly agree. The same below.

The measurement of interactivity (INA) integrated the scales of Chen [53], Dong [54], Yuan [55] and Li [56] with a total of four items (INA1, INA2, INA3 and INA4), such as "This live streaming room shows the packaging and edible method of agricultural products in detail; Hosts can communicate with consumers amiably and answer consumers' questions in a timely and patient manner."

The measurement of presence (PRS) integrated the scales of Ou et al. [23] and Shu [28], with a total of four items (PRS1, PRS2, PRS3 and PRS4), such as "When watching the live-stream, I was very familiar with live streaming scenes (orchards, farmland, markets, plants, etc); When watching the live-stream, I felt that the products were right in front of me."

The measurement of trust (TRU) was based on the scale proposed by Wongkitrungrueng A [57], Liu [58]. There were four measurement items (TRU1, TRU2, TRU3, TRU4 and TRU5),

**Table 1. Distributions characteristics of sample.**

| Items | Classification | All samples | | Valid sample | |
|---|---|---|---|---|---|
| | | N = 1006 | | N = 365 | |
| | | Frequency | % | Frequency | % |
| Gender | Man | 498 | 49.5 | 153 | 41.9 |
| | Woman | 508 | 50.5 | 212 | 58.1 |
| Age | Under 20 years | 132 | 13.1 | 32 | 8.8 |
| | 21–30 years | 238 | 23.7 | 93 | 25.5 |
| | 31–40 years | 204 | 20.3 | 57 | 15.6 |
| | 41–50 years | 289 | 28.7 | 141 | 38.6 |
| | 51 years and older | 143 | 14.2 | 42 | 11.5 |
| Education | Junior high school and below | 150 | 14.9 | 54 | 14.8 |
| | High school/technical school | 373 | 37.1 | 110 | 30.1 |
| | Undergraduate/junior college | 354 | 35.2 | 161 | 44.1 |
| | Postgraduate | 129 | 12.8 | 40 | 11 |
| Occupation | Student | 239 | 23.8 | 76 | 20.8 |
| | Enterprise employees | 250 | 24.9 | 67 | 18.4 |
| | Government agencies/public institutions | 186 | 18.5 | 87 | 23.8 |
| | Freelancer | 167 | 16.6 | 72 | 19.7 |
| | Stay-at-home parents | 31 | 3.1 | 15 | 4.1 |
| | Retired | 21 | 2.1 | 8 | 2.2 |
| | Others | 112 | 11.1 | 40 | 11 |
| Income | 1,000 RMB and below | 118 | 11.7 | 34 | 9.3 |
| | 1,000–4,000 RMB | 435 | 43.2 | 164 | 44.9 |
| | 4,000–7,000 RMB | 235 | 23.4 | 92 | 25.2 |
| | 7,000–10,000 RMB | 117 | 11.6 | 43 | 11.8 |
| | Above 10,000 RMB | 101 | 10.0 | 32 | 8.8 |
| Residence | Urban area | 659 | 65.5 | 219 | 60.0 |
| | Countryside | 347 | 34.5 | 146 | 40.0 |

such as "I believe that the information of the agricultural products on sale were reliable; I believe that the products I received matched the description in the live streaming room."

The measurement of resonance (RES) integrated the scales of Escalas [59], Li [60] and Liu [61] and combined with the characteristics of agricultural live streaming. There were four items in total (RES1, RES2, RES3, and RES4), such as "When listening to the hosts talk about rural life and agricultural production, I felt like I was one of them; I think the stories behind the agricultural production were interesting, and I was very devoted and touched."

The measurement of purchase intention (PUI) integrated the scales of Yu [62] and Sun [24] and revised according to the characteristics of live streaming. There were four measurement items (PUI1, PUI2, PUI3, and PUI4), such as "When watching the live-stream, I successfully placed an order as soon as possible; I am willing to share the live streaming link to my friends as soon as possible."

## Empirical analysis results

### Measurement model

We used SPSS 21 to conduct EFA to test the validity of various measurement items.

The EFA was conducted repeatedly, we removed four items, namely RUS2, RUS3, INA1, and PUI1. The final exploratory factor analysis results are shown in Table 3. All indicator

**Table 2. Variable measurement and source.**

| Latent Constructs | Observed Indicators | References |
|---|---|---|
| Rural Sentiment | 1.I believe that the long-standing and splendid Chinese agricultural culture needs to be passed down. | Ma Deyong, Lu Yizhou (2019) [49] Su Juan, Wang Yan (2020) [50] Yin et al. (2020) [51] Zheng Yi, Zhu Qizhi, Wang Mingzhu (2021) [52] |
|  | 2.Bread is the staff of life. I believe that developing agriculture is a strategic need to ensure the food security of the country. |  |
|  | 3.I think it is important and worthwhile to help farmers earn a good life. |  |
|  | 4.I have a dream of idyllic life in the countryside. |  |
| Interactivity | 1.This live streaming room shows the packaging and edible method of agricultural products in detail. | Chen, C.-C., Lin, Y.-C. (2018) [53] Dong, X., Wang, T. (2018) [54] Yuan Haixia et al. (2022) [55] Li Lianying (2023) [56] |
|  | 2.Hosts can communicate with consumers amiably and answer consumers' questions in a timely and patient manner. |  |
|  | 3.The live streaming room has a variety of interactive activities (such as participating in the screenshot lottery and grabbing red envelopes). |  |
|  | 4.When watching the live-stream, I can communicate the information and content of the agricultural product with other audiences. |  |
| Presence | 1.When watching the live-stream, I was very familiar with live streaming scenes (orchards, farmland, markets, plants, etc). | Ou, C.X., Pavlou, P.A., Davison, R. (2014) [23] Shu Bo, Chen Meidan (2022) [28] |
|  | 2.When watching the live-stream, I felt that I was immersed in the world the hosts had created. |  |
|  | 3.The hosts' descriptions of the agricultural products on sale gave a strong visual impression. |  |
|  | 4.When watching the live-stream, I felt that the products were right in front of me. |  |
| Trust | 1.I believe that the information of the agricultural products on sale were reliable. | Apiradee Wongkitrungruenga, Nuttapol Assarutb (2020) [57] Liu Fengjun et al (2020) [58] |
|  | 2.I believe that the products provided in the live streaming room were guaranteed. |  |
|  | 3.I believe that the products I received matched the description in the live streaming room. |  |
|  | 4.I believe that the quality of the products I purchased from this live streaming room was consistent with my expectations. |  |
|  | 5.I believe that this live streaming room could fulfill the promises made to the consumers. |  |
| Resonance | 1.When listening to the hosts talk about rural life and agricultural production, I felt like these scenes, these people and things were happening around me. | Escalas (2003) [59] Keller (2009) [35] Li Binbin, Long Fei (2021) [60] Liu Ran et al. (2022) [61] |
|  | 2.When listening to the hosts talk about rural life and agricultural production, I felt like I was one of them. |  |
|  | 3.I think the stories behind the agricultural production were interesting, and I was very devoted and touched. |  |
|  | 4.When watching the live-stream, I felt like that I am talking to my old friends about the harvest. |  |
| Purchase Intention | 1.When watching the live-stream, I successfully placed an order as soon as possible. | Yu Kefa (2012) [62] Sun Yuan et al.(2019) [24] |
|  | 2.When watching the live-stream, I added the agricultural production to my shopping cart for comparison and purchase in the future. |  |
|  | 3.I am willing to share the live streaming link to my friends as soon as possible. |  |
|  | 4.When the hosts or farmers mentioned the overstock of agricultural products in the live streaming room, I was willing to place an order to support the farmers. |  |

loadings are almost greater than 0.6 and exceed the cross-loadings, which confirmed discriminant validity. The six factors are highly correlated with a group of items, and it can be considered that each factor corresponds to a variable. The six factors together explained 85.40% of the total variance.

A CFA was dealt with Amos 21.0. The fit indexes of the measurement model were $\chi2/df = 2.854$, RMSEA = 0.071, NFI = 0.945, TLI = 0.956, and CFI = 0.963. These indicators meet the criteria suggested by Bentler and Bonett [63]. According to the results of CFA, the combined reliability of each variable and the average variance extracted are calculated. Meanwhile, Cronbach's α and the correlation coefficient were calculated by SPSS 21.

The test indicators of the scale are presented in Table 4. The Cronbach's α and CR for the six constructs ranged from 0.777 to 0.999 (all above 0.7), having good convergent validity. The

**Table 3. Results of EFA.**

| Items | Factor 1 | Factor 2 | Factor 3 | Factor 4 | Factor5 | Factor 6 |
|---|---|---|---|---|---|---|
| INA2 | 0.277 | **0.642** | 0.231 | 0.43 | 0.152 | 0.218 |
| INA3 | 0.207 | **0.819** | 0.201 | 0.236 | 0.195 | 0.112 |
| INA4 | 0.13 | **0.688** | 0.403 | 0.197 | 0.241 | 0.267 |
| PRS1 | 0.177 | 0.345 | **0.725** | 0.269 | 0.239 | 0.149 |
| PRS2 | 0.216 | 0.215 | **0.743** | 0.28 | 0.289 | 0.264 |
| PRS3 | 0.319 | 0.329 | **0.592** | 0.36 | 0.211 | 0.262 |
| PRS4 | 0.235 | 0.231 | **0.619** | 0.388 | 0.365 | 0.286 |
| TRU1 | 0.258 | 0.222 | 0.327 | **0.686** | 0.301 | 0.242 |
| TRU2 | 0.21 | 0.226 | 0.329 | **0.734** | 0.282 | 0.298 |
| TRU3 | 0.208 | 0.276 | 0.269 | **0.741** | 0.265 | 0.291 |
| TRU4 | 0.178 | 0.261 | 0.268 | **0.738** | 0.335 | 0.251 |
| TRU5 | 0.201 | 0.261 | 0.201 | **0.738** | 0.323 | 0.277 |
| RES1 | 0.253 | 0.245 | 0.341 | 0.437 | **0.625** | 0.238 |
| RES2 | 0.21 | 0.212 | 0.349 | 0.439 | **0.659** | 0.236 |
| RES3 | 0.207 | 0.24 | 0.24 | 0.354 | **0.71** | 0.314 |
| RES4 | 0.207 | 0.222 | 0.305 | 0.327 | **0.668** | 0.382 |
| PUI2 | 0.215 | 0.212 | 0.202 | 0.399 | 0.248 | **0.696** |
| PUI3 | 0.078 | 0.191 | 0.269 | 0.283 | 0.345 | **0.723** |
| PUI4 | 0.39 | 0.208 | 0.266 | 0.351 | 0.246 | **0.577** |
| RUS1 | **0.708** | 0.284 | 0.252 | 0.4 | 0.068 | 0.162 |
| RUS4 | **0.772** | 0.191 | 0.21 | 0.15 | 0.338 | 0.171 |

Note: RUS = Rural Sentiment, INA = Interactivity, PRS = Presence, TRU = Trust, RES = Resonance, PUI = Purchase Intention.

value of AVE was greater than 0.5, indicating that the latent variable had good convergent validity, and the square root of the AVE of each variable was greater than the correlation coefficient, indicating the scale had good discriminative validity [64]. Therefore, the scales had a good performance.

## Empirical results

Table 5 shows the hierarchical regression results for testing the hypothesis.

**Main effect test.** According to Model 8, taking gender, age and income as control variables, interactivity ($\beta = 0.222$, $P < 0.001$) and presence ($\beta = 0.599$, $P < 0.001$) had positive effect on purchase intention significantly. Therefore, H1 and H2 were supported.

**Table 4. Assessment of measurement model.**

| Constructs | Cronbach's α | CR | AVE | RUS | INA | PRS | TRU | RES | PUI |
|---|---|---|---|---|---|---|---|---|---|
| RUS | 0.777 | 0.969 | 0.640 | **0.800**[a] | | | | | |
| INA | 0.873 | 0.986 | 0.694 | 0.661[b] | **0.833** | | | | |
| PRS | 0.930 | 0.995 | 0.770 | 0.699 | 0.775 | **0.877** | | | |
| TRU | 0.967 | 0.999 | 0.857 | 0.691 | 0.739 | 0.805 | **0.926** | | |
| RES | 0.957 | 0.998 | 0.847 | 0.674 | 0.709 | 0.820 | 0.842 | **0.920** | |
| PUI | 0.872 | 0.987 | 0.698 | 0.654 | 0.682 | 0.766 | 0.810 | 0.818 | **0.836** |

Note: CR: composite reliability, AVE: average variance extracted.

[a] Diagonal elements (bold values) represent the square root of AVE.

[b] The elements below the diagonal represent the correlation coefficients.

**Table 5. Hierarchical regression results for hypothesis testing.**

| Variable | Trust | | | Resonance | | | Purchase Intention | | | |
|---|---|---|---|---|---|---|---|---|---|---|
| | Model 1 | Model 2 | Model 3 | Model 4 | Model 5 | Model 6 | Model7 | Model 8 | Model 9 | Model 10 |
| **1.Control variable** | | | | | | | | | | |
| Gender | 0.292* | 0.252** | 0.282** | 0.205 | 0.166* | 0.187 | 0.208 | 0.169 | 0.204** | 0.043 |
| Age | -0.124* | -0.08** | -0.077** | -0.001 | 0.032 | 0.038 | -0.042 | -0.005 | -0.001 | 0.01 |
| Income | 0.069 | 0.031 | 0.024 | 0.049 | 0.005 | -0.002 | 0.035 | -0.004 | -0.004 | -0.011 |
| **2. Independent Variable** | | | | | | | | | | |
| INA | | 0.27*** | 0.179*** | | 0.19*** | 0.111** | | 0.222*** | 0.133** | 0.034 |
| PRS | | 0.585*** | 0.494*** | | 0.666*** | 0.594*** | | 0.599*** | 0.511*** | 0.128** |
| **3.Moderator** | | | | | | | | | | |
| RUS | | | 0.178*** | | | 0.135** | | | 0.155** | 0.047 |
| RUS×INA | | | -0.066** | | | -0.069** | | | -0.042 | 0.005 |
| RUS×PRS | | | 0.049 | | | 0.05 | | | 0.016 | -0.018 |
| **4.Mediator** | | | | | | | | | | |
| TRU | | | | | | | | | | 0.322*** |
| RES | | | | | | | | | | 0.376*** |
| $R^2$ | 0.02 | 0.692 | 0.719 | 0.005 | 0.69 | 0.709 | 0.006 | 0.609 | 0.632 | 0.732 |
| $\Delta R^2$ | 0.02 | 0.672 | 0.027 | 0.005 | 0.685 | 0.019 | 0.006 | 0.603 | 0.023 | 0.1 |
| F | 2.449* | 161.62*** | 114.00*** | 0.591 | 159.60*** | 108.51*** | 0.7 | 111.89*** | 76.49*** | 96.64*** |
| $\Delta F$ | 2.449* | 392.41*** | 11.34*** | 0.591 | 396.18*** | 7.939*** | 0.7 | 277.08*** | 7.445*** | 65.82*** |

Note: N = 365

* p<0.1

** p<0.05

*** p<0.001. The same below.

**Mediating effect test.** The hypotheses were tested by hierarchical regression analysis which was proposed by Baron and Kenny [65]. According to Model 2, the regression coefficient of interactivity ($\beta = 0.27$, P < 0.001) and presence ($\beta = 0.585$, P < 0.001) were significant. Meanwhile Model 5 shows that the regression coefficient of the interactivity ($\beta = 0.19$, P < 0.001) and presence ($\beta = 0.666$, P < 0.001) were significant. In Model 10, trust ($\beta = 0.322$, P < 0.001) and resonance ($\beta = 0.376$, P < 0.001) were significantly correlated with purchase intention. After mediation variables trust and resonance were added to Model 9, the coefficient of interactivity on purchase intention ($\beta = 0.034$, n.s.) became insignificant and presence on purchase intention ($\beta = 0.128$, P < 0.05) became smaller, H3a, H3b, H4a and H4b were supported.

Following the procedure of Hayes and Preacher [66], we use the Sobel test and bootstrapping method to further confirm the mediating effect, the four indirect effect paths were significant in Table 6, the Z-values of Sobel test range from 3.363 to 5.834, all p-value are less than 0.01.

**Table 6. Sobel test calculation of mediating effect.**

| Indirect effect path | Estimate | S.E. | Mediator effect | Z-value | p-value |
|---|---|---|---|---|---|
| INA→TRU→PUI | a = 0.267<br>b = 0.322 | Sa = 0.048<br>Sb = 0.060 | ab = 0.086 | 3.862 | 0.000 |
| INA→RES→PUI | a = 0.190<br>b = 0.376 | Sa = 0.048<br>Sb = 0.059 | ab = 0.071 | 3.363 | 0.001 |
| PRS→TRU→PUI | a = 0.588<br>b = 0.322 | Sa = 0.046<br>Sb = 0.060 | ab = 0.189 | 4.948 | 0.000 |
| PRS→RES→PUI | a = 0.667<br>b = 0.376 | Sa = 0.046<br>Sb = 0.059 | ab = 0.251 | 5.834 | 0.000 |

**Table 7. Bootstrapping test of mediating effects.**

| Mediator effect pathway | Effect | S.E. | BootLL95%CI | BootUL95%CI |
|---|---|---|---|---|
| INA→TRU→PUI | 0.058 | 0.022 | 0.024 | 0.114 |
| INA→RES→PUI | 0.042 | 0.023 | 0.004 | 0.095 |
| PRS→TRU→PUI | 0.159 | 0.047 | 0.082 | 0.274 |
| PRS→RES→PUI | 0.223 | 0.054 | 0.118 | 0.328 |

Note: Bootstrapping sample size 2000.

The bootstrapping test (using the Mplus7.4 program, sampling size 2000) was used to clarify the indirect effects. Shrout and Bolger [67] proposed that if 0 is not included in the CI, the indirect effect can be considered as different from 0. Table 7 shows that trust had a significant mediating effect on the effects of interactivity (LL95%CI = 0.024, UL95%CI = 0.114, excluding 0) and presence (LL95%CI = 0.082, UL95%CI = 0.274, excluding 0) on purchase intention, while resonance had a significant mediating effect on interactivity (LL95%CI = 0.004, UL95%CI = 0.095, excluding 0) and presence (LL95%CI = 0.118, UL95%CI = 0.328, excluding 0) on the influence of purchase intention, which indicates that the indirect effect was statistically significant and further supports H3a, H3b, H4a and H4b.

**Moderating effect test.** The cross-term of rural sentiment and interactivity was significant ($\beta = -0.066$, $P < 0.05$) in Model 3, but the cross-term of rural sentiment and presence was insignificant ($\beta = 0.049$, n.s.). The cross-term of rural sentiment and interactivity was significant ($\beta = -0.069$, $P < 0.05$) in Model 6, but the cross-term of rural sentiment and presence was insignificant ($\beta = 0.05$, n.s.). The results are inconsistent with H5, but the significant non zero coefficient of the cross term suggests that we should reconsider the moderate direction of rural sentiment. After rural sentiment was added to Model 2, the coefficient of interactivity decreased from 0.27 to 0.179, the coefficient of presence decreased from 0.585 to 0.494 in Model 3, and the coefficient of rural sentiment was 0.178 and significant, demonstrating rural sentiment has a certain substitute effect on interactivity and presence when predicting trust. A negative coefficient of cross term indicates a negative moderating effect, meaning that respondents with intense rural sentiment are more likely to trust the live streaming room, while less interactivity with the host is needed. Just like a student who is willing to participate in class can achieve better learning effect instead of activating the classroom atmosphere by the teacher. The change from Model 5 to Model 6 is similar.

Bootstrapping test of the moderating effect is shown in Table 8. Under low level of rural sentiment, the effect of interactivity on resonance is higher (effect value is 0.220, $p<0.05$). Under high level of rural sentiment, the effect of interactivity on resonance is lower(effect value is 0.001, n.s.). The difference of the effects is -0.219, $p < 0.1$, the 90% confidence interval is [-0.420, -0.020], excluding 0. The difference in the moderating effect of the other three paths on the high and low levels of rural sentiment is not significant.

Fig 2 illustrates the moderating effect of rural sentiment.

**Moderated mediation test.** Following Edwards and Lambert, bootstrapping method (using the Mplus7.4 program, sampling size 2000) was used to test whether rural sentiment moderates the indirect effect [68]. According to Table 9, under high level of rural sentiment, the path INA→RES→PUI is insignificant, under low level of rural sentiment, the path is 0.083 and significant, the difference of the effects is significant, with a 95% confidence interval of [-0.199, -0.001]. In other words, the higher the degree of rural sentiment, the weaker interactivity through resonance to purchase intention. The results of Table 9 are consistent with

**Table 8. Bootstrapping test of moderating effects.**

| effect path | Rural Sentiment | Effect | S.E. | P-value | BootLL90%CI | BootUL90%CI |
|---|---|---|---|---|---|---|
| INA→TRU | High | 0.074 | 0.089 | 0.407 | -0.075 | 0.222 |
| | Low | 0.284** | 0.086 | 0.001 | 0.146 | 0.431 |
| | Difference | -0.210 | 0.132 | 0.111 | -0.428 | 0.009 |
| INA→RES | High | 0.001 | 0.081 | 0.988 | -0.131 | 0.131 |
| | Low | 0.220** | 0.084 | 0.009 | 0.080 | 0.361 |
| | Difference | -0.219* | 0.122 | 0.072 | -0.420 | -0.020 |
| PRS→TRU | High | 0.571*** | 0.092 | 0.000 | 0.417 | 0.719 |
| | Low | 0.416*** | 0.100 | 0.000 | 0.251 | 0.580 |
| | Difference | 0.155 | 0.142 | 0.274 | -0.085 | 0.383 |
| PRS→RES | High | 0.673*** | 0.081 | 0.000 | 0.535 | 0.801 |
| | Low | 0.515*** | 0.088 | 0.000 | 0.363 | 0.653 |
| | Difference | 0.158 | 0.123 | 0.198 | -0.038 | 0.361 |

Note: Bootstrapping sample size 2000.

Table 8. H6 were also not support. However, the negative moderate effect value is still a beneficial finding.

## Heterogeneity analysis

Considering that cultural backgrounds may lead to differences in consumer emotional responses, this study further examines whether there is heterogeneity in consumer purchase

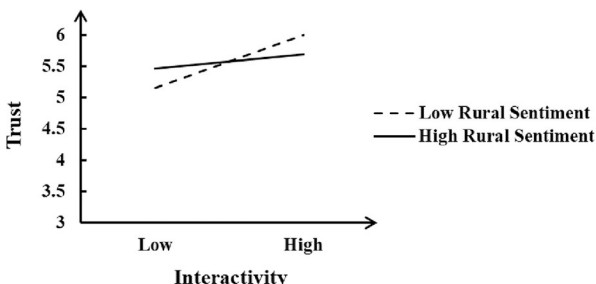

a.The moderating effect of rural sentiment on the relationship between interactivity and trust

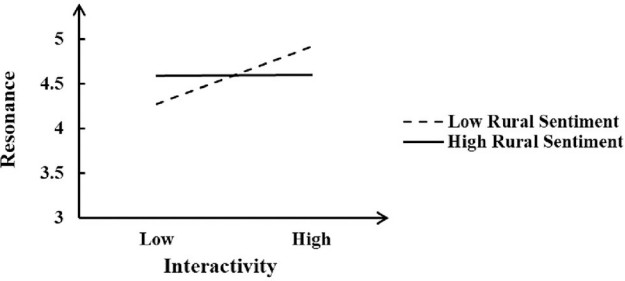

b.The moderating effect of rural sentiment on the relationship between interactivity and resonance

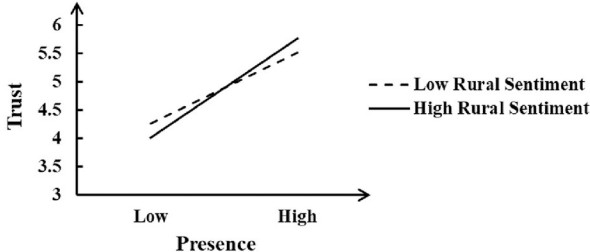

c.The moderating effect of rural sentiment on the relationship between presence and trust

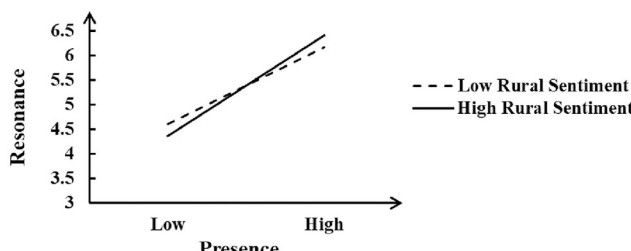

d.The moderating effect of rural sentiment on the relationship between presence and resonance

**Fig 2. The moderating effect of rural sentiment.**

**Table 9. Bootstrapping test of moderated mediation effects.**

| effect path | Rural Sentiment | Effect | S.E. | P-value | BootLL95%CI | BootUL95%CI |
|---|---|---|---|---|---|---|
| INA→TRU→PUI | High | 0.024 | 0.029 | 0.411 | -0.027 | 0.089 |
| | Low | 0.092** | 0.034 | 0.007 | 0.037 | 0.178 |
| | Difference | -0.068 | 0.045 | 0.132 | -0.177 | 0.005 |
| INA→RES→PUI | High | 0.000 | 0.032 | 0.988 | -0.062 | 0.064 |
| | Low | 0.083** | 0.036 | 0.023 | 0.024 | 0.168 |
| | Difference | -0.082 | 0.050 | 0.101 | -0.199 | -0.001 |
| PRS→TRU→PUI | High | 0.184** | 0.056 | 0.001 | 0.094 | 0.322 |
| | Low | 0.134** | 0.048 | 0.005 | 0.055 | 0.259 |
| | Difference | 0.050 | 0.047 | 0.284 | -0.029 | 0.167 |
| PRS→RES→PUI | High | 0.253*** | 0.063 | 0.000 | 0.134 | 0.378 |
| | Low | 0.194** | 0.056 | 0.001 | 0.094 | 0.313 |
| | Difference | 0.059 | 0.050 | 0.235 | -0.025 | 0.173 |

Note: Bootstrapping sample size 2000.

intention in live streaming rooms between urban and rural areas. This study divided the respondents into urban and rural groups and conducted regression analysis separately.

The coefficient from rural sentiment to trust in the urban group is not significant, while the coefficient from rural sentiment to trust in the rural group is 0.293, which is statistically significant. The p-value was 0.049 and significant through the seemingly unrelated test [69], confirming the difference between groups of urban and rural is statistically significant. For the rural group, a higher level of rural sentiment has intense connection with trust in the live streaming room. A possible explanation is that the respondents in the rural group have real rural life experiences, so they could identify the value and quality of agricultural products sold in the live streaming room accurately. Table 10 shows that there are seven differences between groups of urban and rural. Thus, the comparison of group regression has certain research value. Due to the limit of the data we have obtained, we only conducted a comparison of urban and rural groups, and other background differences are worth further testing.

## Conclusions

This study explored the influence of emotional factors in agricultural products live streaming and drew the following conclusions.

The results of Model 8 indicate that interactivity and presence had a positive impact on consumers' purchase intention. The hosts can talk more about the production scenarios, processes of agricultural products and interesting stories in rural life, as well as communicating with consumers amiably and in real-time. In addition, consumers are encouraged to interact by sending a barrage of questions, not only increasing the freshness of the live streaming but also enhancing consumer fascination, shortening the distance between the products and consumers, and stimulating their enthusiasm and potential consumption desire.

The results of Models 2 and 5 indicate that interactivity and presence positively correlated with trust and resonance. Meanwhile Model 10 shows that trust and resonance positively correlated with purchase intention. That is to say, interactivity and presence had an impact on purchase intention via trust and resonance. The indirect path was verified by the results of bootstrapping test and Sobel test. We consider that consumers' trust and resonance are very important emotional factors in live streaming of agricultural products, which stimulate consumers' enthusiastic purchase intention and purchase behavior.

**Table 10. Differences between urban area and countryside group regression results.**

| Path | Group 1 Urban area | Group 2 Countryside | p-value |
|---|---|---|---|
| INA→TRU | 0.185** | 0.209** | 0.843 |
| PRS→TRU | 0.495*** | 0.475*** | 0.861 |
| RUS→TRU | 0.072 | 0.293*** | 0.049** |
| RUS×INA→TRU | -0.118** | 0.021 | 0.061* |
| RUS×PRS→TRU | 0.082 | -0.014 | 0.226 |
| INA→RES | 0.075 | 0.347** | 0.027** |
| PRS→RES | 0.624*** | 0.427*** | 0.098* |
| RUS→RES | 0.062 | 0.211** | 0.110 |
| RUS×INA→RES | -0.135** | 0.082* | 0.000*** |
| RUS×PRS→RES | 0.112** | -0.083** | 0.002** |
| INA→PUI | -0.013 | 0.101 | 0.383 |
| PRS→PUI | 0.201** | 0.052 | 0.278 |
| RUS→PUI | 0.007 | 0.145** | 0.198 |
| RUS×INA→PUI | 0.050 | -0.037 | 0.186 |
| RUS×PRS→PUI | -0.048 | 0.008 | 0.441 |
| TRU→PUI | 0.427*** | 0.112 | 0.029** |
| RES→PUI | 0.333** | 0.441*** | 0.502 |

Note: The p-value is obtained by using the seemingly uncorrelated model SUR to test the significance of coefficient differences among explanatory variables in different groups.

Contrary to our expectations, a negative moderating effect of rural sentiment has been founded, rural sentiment has a negative moderating effect on the relationship between interactivity and trust as shown in Model 3. Rural sentiment has a moderating effect as shown in Model 6. It was also found that given China's long agricultural civilization, the deep connection to farming has influenced everyone over the years. Rural sentiment was also reflected in the nostalgia caused by rural experience or consumers yearning for the countryside and respecting farmers because of their love for nature. Moreover, live streaming marketing was found to use an interactive way of telling stories and feelings, inspiring the emotional response of consumers, heightening consumers' recognition of the streaming room; this could be of great value to effective communication in the streaming room.

The difference of indirect effect of interactivity on purchase intention through resonance under high and low levels of rural sentiment is significant. It can be seen that the higher the degree of rural sentiment, the weaker interactivity through resonance to purchase intention.

Differences of regression results between urban and rural group indicate that the different cultural backgrounds of consumers have a different impact on their emotional responses in live streaming of agricultural products. The rural sentiment has a stronger explanatory power on trust, resonance and purchase intention in the group of consumers who lived in countryside.

## Research contribution

This paper studies how emotional factors impact the consumers' purchase intention in live steaming of agricultural products. We constructed a moderated mediation model of the interaction, presence, trust, resonance and purchase intention under rural sentiment. This paper makes four main contributions.

Firstly, this paper enriches Consumer Behavior Theory by testing the interaction and integration effect of emotional factors in live streaming of agricultural products. In contrast to

prior literature that focused on single emotional factor's effect on purchase intention or consumers behaviors, this paper integrates four emotional factors organically, testing and verifying the main effects of presence, the mediating effects of trust and resonance, and the moderating effects of rural sentiment on purchase intention. The integrated effects of the four emotional factors provide a more specific description of consumers' psychological activities in live streaming, thereby enriching Consumer Behavior Pattern theory.

Secondly, this paper contributes to live streaming of agricultural products by introducing rural sentiment, which has a incremental value for live streaming research. Rural sentiment, as a special emotion with Chinese characteristics, plays an important role in agricultural product marketing. Although, less literature paid attention to rural sentiment in live streaming. In this paper, we find positive direct effects of rural sentiment on trust and resonance, and negative moderating effects, which is inconsistent with Su Juan [50]. In their study, the moderating effect of homesickness(the concept closest to the rural sentiment in literature) was positive. However, the impact of homesickness on satisfaction is not significant. In this study, the rural sentiment has a significant positive impact on trust and resonance, indicating that rural sentiment has a certain substitute effect on interactivity and presence. The finding inspires scholars to analyze live streaming of agricultural products from sentimental perspective.

Thirdly, this paper develops adjusted measurement scales of resonance and rural sentiment with high reliability and validity. Considering the characteristics of live streaming of agricultural products, we incorporate elements and emotions related to traditional Chinese agricultural culture, forming more specific scales reflecting the characteristics of Chinese countryside, farmers, and agriculture. This adjusted measurement scales of resonance and rural sentiment might provide a more valuable measurement tool for emotional factors in live streaming research.

In addition, a cross-cultural test is conducted by heterogeneity analysis, recognizing that urban and rural residents respond differently to rural sentiment. Rural sentiment has a stronger explanatory power on trust, resonance and purchase intention in the group of consumers who lived in countryside than in cities. This finding might give a sign to live streaming managers to improving their marketing strategy.

## Marketing enlightenment

The following marketing enlightenment is proposed in this paper.

Firstly, frequent interactions should be encouraged to enhance the consumers' intention to purchase. Compared to short video, the live streaming should focus on mobilizing the atmosphere of real-time interactivity, including the communication between hosts and users and the communication among users. The hosts might actively encourage consumers to raise questions through bullet screen, and share the thoughts of agricultural products with other consumers. The scene of the live-stream room must be designed and set up carefully, so as to improve consumers' presence, and provide consumers with an immersive shopping experience.

Secondly, information about the products on sale should be provided adequately in the live streaming room, product delivered should match the consumers' expectations, thereby enhance consumers' trust. At the same time, hosts might chat with consumers like friends. For example, the host can tells his or her wonderful experiences in the countryside, stimulate consumers' emotional response, and affect their purchase intention. Hosts might also describe the scene of farmers' farming or stream live in farmland, which make consumers trust the quality of agricultural products and be very willing to assist farmers.

Thirdly, arousing consumers' rural sentiment and motivating consumers' resonance are necessary to strengthen consumers' interactive behaviors in live marketing room. Rural

sentiment is reflected in the nostalgic emotions caused by rural experiences or consumers longing for the countryside and respecting farmers due to their love for nature. Hosts might describe the rural life, personal rural experience, and stories and history to arouse consumers' rural sentiment and resonance.

Fourthly, local government might play a special role in live streaming marketing. Compared with the general hosts, local government officials are more credible. Government officials, acting as "specially invited guest and assistant" in the live streaming room, will make consumers more confident in the products. Specifically, county officials living in countryside might introduce the agricultural products more comprehensively than farmer hosts. Meanwhile, they are supposed to propagate the regional history, culture, and social customs, which will greatly increase the attractiveness of the region and products.

## Deficiency and prospect

There are a few deficiencies in our study. Firstly, the moderating role of rural sentiment is not consistent with our expectations. The possible cause is that the indicators in the scale (such as items 2 and 3) characterize in the macro level, so that most of the respondents choose "6" or "7", which may cause smaller changes in variables and less explanatory power. To obtain the real emotional reactions and verify the moderating role of rural sentiment, the future research will design the scale questions from a micro perspective.

Secondly, this article mainly obtained samples through e-questionnaires, and the samples are not representative enough, so it is necessary to further use various methods to obtain samples (such as in-depth interviews and focus group discussions) and gain a deeper understanding of consumers' emotional responses.

Thirdly, this article had examined the immediate emotional response of consumers in live streaming, but the deficiency is that there is no longitudinal study, and no follow-up tracking of consumer behavior is carried out. Therefore, this subsequent study will consider capturing changes in consumer behavior over time to gain insight into the long-term impact of emotional factors on purchase decisions amid evolving market trends and technological advancement.

## Supporting information

**S1 File. Questionnaire-Chinese original edition.**
(DOCX)

**S2 File. Questionnaire-English copy edition.**
(DOCX)

**S3 File. Data-Chinese original edition.**
(XLSX)

**S4 File. Data-English copy edition.**
(XLSX)

## Acknowledgments

We are grateful to all the participants for their contributions to this study. We would also like to thank the editors and reviewers for their suggestions.

## Author Contributions

**Conceptualization:** Tianming Han.

**Data curation:** Tianming Han.

**Formal analysis:** Jing Han, Weibao Li.

**Investigation:** Jing Han, Weibao Li.

**Methodology:** Tianming Han, Weibao Li.

**Resources:** Jing Han, Jia Liu.

**Software:** Tianming Han, Jia Liu.

**Supervision:** Weibao Li.

**Writing – original draft:** Tianming Han, Jing Han.

**Writing – review & editing:** Jia Liu, Weibao Li.

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
