## [Decision Letter · Decision Letter 0]

13 Nov 2023

PONE-D-23-29134Effect of emotional factors on purchase intention in streaming live marketing of agricultural products: a moderated mediation modelPLOS ONE

Dear Dr. Li,

Thank you for submitting your manuscript to PLOS ONE. After careful consideration, we feel that it has merit but does not fully meet PLOS ONE’s publication criteria as it currently stands. Therefore, we invite you to submit a revised version of the manuscript that addresses the points raised during the review process.

We look forward to receiving your revised manuscript.

Kind regards,

Mingyue Fan, Ph.D.

Academic Editor

PLOS ONE

2. PLOS requires an ORCID iD for the corresponding author in Editorial Manager on papers submitted after December 6th, 2016. Please ensure that you have an ORCID iD and that it is validated in Editorial Manager. To do this, go to ‘Update my Information’ (in the upper left-hand corner of the main menu), and click on the Fetch/Validate link next to the ORCID field. This will take you to the ORCID site and allow you to create a new iD or authenticate a pre-existing iD in Editorial Manager. Please see the following video for instructions on linking an ORCID iD to your Editorial Manager account: https://www.youtube.com/watch?v=_xcclfuvtxQ.

3. Please amend your authorship list in your manuscript file to include authors Weibao Li, Tianming Han, Jing Han, and Jia Liu.

Additional Editor Comments:

There are several points that authors can revise:

1. Introduction: The necessity of research content is not reflected, and it does not indicate that the research content has not been studied by previous researchers or lacks sufficient research. The first two sections lack some logical coherence, and are suggested to be made some adjustments.

2. Literature Review: The theory used in this study, the S-O-R (Stimulus-Organism-Response) theory, lacks a certain level of innovation.

3. Method: Based on what criteria are the data from 3-4 questions integrated into a latent variable, and there being a lack of relevant explanations and citation references for the combination of data sources.

4. Conclusion and Discussion: In this section, it is suggested to reorganize the content by comparing and discussing one's own conclusions with previous research, and presenting valuable viewpoints.

5. Reference: Literature is scarce, and some of it is outdated. It is recommended to appropriately increase and update the references.

6. Please get the paper read by a professional language editor.

7. Run the article on a plagiarism check software like Turnitin and eliminate similarities from other publications. The Similarity Index should be within 10 percent.

Please take a look at the reviewers' comments and address them. Authors can ignore the citations listed in the reviewers' comments if they are irrelevant.

I would also invite the authors to highlight the unique contribution to theory and methodology in the revised version. Following papers will be helpful during revision.

Leidner, D. E. (2020). What's in a Contribution?. Journal of the Association for Information Systems, 21(1), 2.

Whetten, D. A. (1989). What constitutes a theoretical contribution?. Academy of Management Review, 14(4), 490-495.

Bergh, D. D., Boyd, B. K., Byron, K., Gove, S., & Ketchen Jr, D. J. (2022). What constitutes a methodological contribution?. Journal of Management, 48(7), 1835-1848.

Reviewers' comments:

Reviewer's Responses to Questions

**Comments to the Author**

1. Is the manuscript technically sound, and do the data support the conclusions?

Reviewer #1: No

Reviewer #2: Partly

2. Has the statistical analysis been performed appropriately and rigorously? 

Reviewer #1: No

Reviewer #2: No

3. Have the authors made all data underlying the findings in their manuscript fully available?

Reviewer #1: Yes

Reviewer #2: Yes

4. Is the manuscript presented in an intelligible fashion and written in standard English?

Reviewer #1: Yes

Reviewer #2: Yes

5. Review Comments to the Author

Reviewer #1: The paper studies how emotional factors impact the consumers’ purchase intention in live steaming. They constructed a moderated mediation model of the interaction, resonance and purchase intention under rural sentiment. They found several interesting findings. Overall, the paper examines a timely and promising topic. My concerns for the paper is as follows:

1. The contribution of the paper. The hypotheses developed in the paper is well expected. For example, H1 proposed that interaction in streaming live marketing of agricultural products has a positive impact on consumers’ purchase intention. The relationship between interaction and purchase intention has been studied by the literature. Therefore, the incremental value of the current study is not clear. I suggest the authors revise the framework.

2. The literature review is too brief. The authors mainly discuss about stimulus–Organism–Response (SOR) in the literature review but ignored the literature on live streaming. I suggest the authors add the literature on live streaming to the paper.

3. The theoretical support for the survey questions. The literature to support the development of the survey questions seems not very solid. I suggest the authors develop the survey questions using high-cited articles.

I hope the authors find the suggestions useful. Good luck!

Reviewer #2: Thank you for the Editor's invitation to review manuscript PONE-D-23-29134, “Effect of emotional factors on purchase intention in streaming live marketing of agricultural products: a moderated mediation model,” about the potential to strengthen its current findings and contribute to a more thorough understanding of the role of emotional factors in agricultural product marketing within the context of live streaming. While the provided article comprehensively explores the impact of emotional factors on consumers' purchase decisions during the live streaming of agricultural products, certain confines should be acknowledged, and a chance should be given to the authors to redo their work after MAJOR REVISION. Here are some suggestions to improve the article and please cite new research it’s related to author’s study below:

Consider enlarging the sample size and diversifying demographic representation to enhance the external validity of findings, providing more nuanced insights into how emotional factors influence purchase intention across various consumer groups. Explore the feasibility of conducting a cross-cultural analysis, recognizing that different cultures may respond differently to emotional cues in marketing. This could offer valuable insights into the universality or cultural specificity of emotional factors in agricultural product marketing. Assess the feasibility of a longitudinal study to capture changes in consumer behavior over time, providing insights into the long-term impact of emotional factors on purchase decisions amid evolving market trends and technological advancements. Incorporate qualitative methods, such as in-depth interviews or focus group discussions, to gain a deeper understanding of consumers' emotional responses. Qualitative data can complement quantitative findings, offering richer insights into the nuances of emotional experiences during live streaming. Expand the scope of emotional factors studied beyond interaction, resonance, and rural sentiment. Consider exploring additional emotional factors like trust, excitement, or nostalgia that may play a role in agricultural product marketing, providing a more comprehensive Please discuss the potential policy implications arising from the findings. For example, if emotional factors significantly impact agricultural product sales, policymakers may consider supporting initiatives that enhance emotional engagement in the agricultural sector. Provide practical recommendations for marketers based on the study's findings. Offer insights into how agricultural product marketers can leverage emotional factors in their live streaming strategies, presenting actionable recommendations that industry professionals can apply.

Good Luck

6. PLOS authors have the option to publish the peer review history of their article (what does this mean?). If published, this will include your full peer review and any attached files.

Reviewer #1: No

Reviewer #2: No

---

## [Author Response · Author response to Decision Letter 0]

15 Jan 2024

PONE-D-23-29134

Effect of emotional factors on purchase intention in streaming live marketing of agricultural products: a moderated mediation model

PLOS ONE

Dear editor,

We sincerely thank you and all reviewers for your valuable comments that we have used to improve the quality of our manuscript. 

According to these valuable comments, we made several major revisions to our manuscript. First, by extensive literature review, we added "presence" as an additional independent variable and "trust" as an additional mediating variable, and analysed the relationship between new and original variables; Second, we revised the survey questionnaire, conducted a new survey using quota sampling method, and increased the sample size; Third, we reanalyzed the data obtained from the survey questionnaire and drew richer and more specific results; Fourth, corresponding conclusions and research contrubiton were conducted at the end of the manucript. 

For specific revision and response, please refer to the rebattle letter.

We look forward to hearing from you regarding our submission. We would be glad to respond to any further questions and comments that you may have.

Your sincerely

Weibao Li

Jan 15 2024

---

## [Decision Letter · Decision Letter 1]

24 Jan 2024

Effect of emotional factors on purchase intention in live streaming  marketing of agricultural products: a moderated mediation model

PONE-D-23-29134R1

Dear Dr. Li,

We’re pleased to inform you that your manuscript has been judged scientifically suitable for publication and will be formally accepted for publication once it meets all outstanding technical requirements.

Kind regards,

Mingyue Fan, Ph.D.

Academic Editor

PLOS ONE

**Comments to the Author**

1. If the authors have adequately addressed your comments raised in a previous round of review and you feel that this manuscript is now acceptable for publication, you may indicate that here to bypass the “Comments to the Author” section, enter your conflict of interest statement in the “Confidential to Editor” section, and submit your "Accept" recommendation.

Reviewer #1: All comments have been addressed

Reviewer #2: All comments have been addressed

2. Is the manuscript technically sound, and do the data support the conclusions?

Reviewer #1: Yes

Reviewer #2: Yes

3. Has the statistical analysis been performed appropriately and rigorously? 

Reviewer #1: Yes

Reviewer #2: Yes

4. Have the authors made all data underlying the findings in their manuscript fully available?

Reviewer #1: Yes

Reviewer #2: Yes

5. Is the manuscript presented in an intelligible fashion and written in standard English?

Reviewer #1: Yes

Reviewer #2: Yes

6. Review Comments to the Author

Reviewer #1: The authors have addressed all my concerns. I very much appreciate it. I suggest accept the paper in the current form.

Reviewer #2: Thank you to Authors, and i am very happy to see many to improve paper. All my suggestions have been revised and addressed well

7. PLOS authors have the option to publish the peer review history of their article (what does this mean?). If published, this will include your full peer review and any attached files.

Reviewer #1: No

Reviewer #2: **Yes: **Muhammad Bilawal Khaskheli

---

## [Editor Report · Acceptance letter]

19 Feb 2024

PONE-D-23-29134R1 

PLOS ONE

Dear Dr. Li, 

I'm pleased to inform you that your manuscript has been deemed suitable for publication in PLOS ONE. Congratulations! Your manuscript is now being handed over to our production team.

Kind regards, 

on behalf of

Dr. Mingyue Fan 

Academic Editor

PLOS ONE